# Predicting the Dynamic Parameters of Multiphase Flow in CFD (Dam-Break Simulation) Using Artificial Intelligence-(Cascading Deployment)

**S. Sina Hosseini Boosari**

Department of Petroleum and Natural Gas Engineering, West Virginia University, Morgantown, WV 26505, USA; sehosseiniboosari@mix.wvu.edu

**Abstract:** Multiphase flow of oil, gas, and water occurs in a reservoir's underground formation and also within the associated downstream pipeline and structures. Computer simulations of such phenomena are essential in order to achieve the behavior of parameters including but not limited to evolution of phase fractions, temperature, velocity, pressure, and flow regimes. However, within the oil and gas industry, due to the highly complex nature of such phenomena seen in unconventional assets, an accurate and fast calculation of the aforementioned parameters has not been successful using numerical simulation techniques, i.e., computational fluid dynamic (CFD). In this study, a fast-track data-driven method based on artificial intelligence (AI) is designed, applied, and investigated in one of the most well-known multiphase flow problems. This problem is a two-dimensional dam-break that consists of a rectangular tank with the fluid column at the left side of the tank behind the gate. Initially, the gate is opened, which leads to the collapse of the column of fluid and generates a complex flow structure, including water and captured bubbles. The necessary data were obtained from the experience and partially used in our fast-track data-driven model. We built our models using Levenberg Marquardt algorithm in a feed-forward back propagation technique. We combined our model with stochastic optimization in a way that it decreased the absolute error accumulated in following time-steps compared to numerical computation. First, we observed that our models predicted the dynamic behavior of multiphase flow at each time-step with higher speed, and hence lowered the run time when compared to the CFD numerical simulation. To be exact, the computations of our models were more than one hundred times faster than the CFD model, an order of 8 h to minutes using our models. Second, the accuracy of our predictions was within the limit of 10% in cascading condition compared to the numerical simulation. This was acceptable considering its application in underground formations with highly complex fluid flow phenomena. Our models help all engineering aspects of the oil and gas industry from drilling and well design to the future prediction of an efficient production.

**Keywords:** computational fluid dynamic (CFD); dam-break; multiphase flow simulation; artificial neural network (ANN)

---

## 1. Introduction

In different types of hydrocarbon reservoirs, oil and gas components are produced constantly and combined with solid particles and water. The $CO_2$ injection process, the separation of mixture components, and the transportation by pipes from reservoirs to the surface are some of the multiphase flows in the oil and gas industry [1,2]. Therefore, it is important in petroleum engineering to know how the multiphase flow is treated and how their variables could be predicted [3]. Modeling and numerical simulations of unconventional reservoirs are much more sophisticated in comparison to the conventional reservoir modeling because of their complex flow characteristics [4].

Reducing the computational time for the fluid flow simulations by developing a smart proxy model (SPM) is the main objective of this research. To achieve this, artificial intelligence (AI) and machine learning techniques were employed to build and design SPMs that possessed the same precision as CFD simulations. The main computed data were generated by the OpenFOAM simulator program. One of the disadvantages of the all multiphase flow simulator is that it is time-consuming. Even for the simple simulation, engineers have to wait a long time to receive results. Therefore, we used the SPM as a new method to overcoming this shortage.

## 2. Literature Review

Most of the current available techniques using AI in petroleum science are related to problems that do not contain significant dynamic characteristics. Predicting the multiphase flow parameters generated by CFD requires a different technique and has gained importance in recent years. The first success of AI usage in petroleum science was related to utilizing an artificial neural network (ANN) to design a model for predicting the formation permeability [5]. In [6], authors introduced ANN as a new technique for predicting gas storage and post-fracture well performance. Most recent studies have focused on dynamic properties, such as: applying the AI model to predict the parameters of gas-solid flow in gasification problems simulated by MFiX (version 17.2, National Energy Technology Laboratory (NETL), Pittsburgh, PA, USA) [7]; attempting to achieve a new correlation by estimating the crude oil viscosity with the neural network (NN) model [8]; oil reservoir parameter predictions using data-driven proxy modeling techniques [9]; production performance prediction, such as cumulative steam-oil ratio (SOR), cumulative oil production, oil production rate, and oil recovery factor using a radial basis function network proxy [10]; predicting the velocity profile and temperature by applying ANN to a CFD simulator with the claim that the model is able to predict temperature and tile flow rate with an average error of <0.6 °C and 0.7%, respectively [11].

## 3. OpenFOAM Numerical Solution

In the CFD simulations using the OpenFOAM package (version 5, The OpenFOAM Foundation Ltd, London, UK), physical properties such as pressure, velocity, and phase fraction at each time-step were the main unknowns (Figure 1).

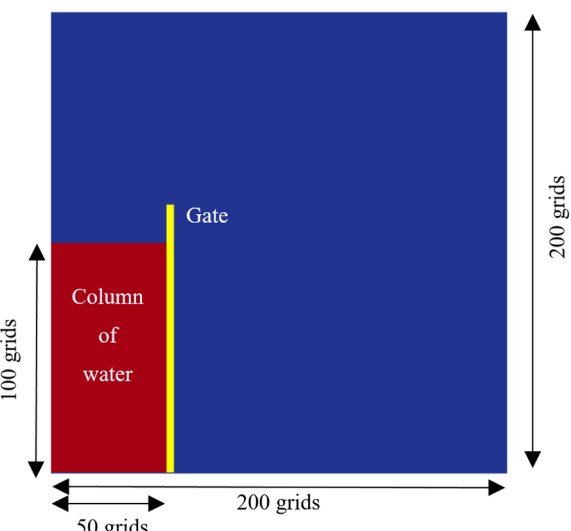

**Figure 1.** Schematic view of initial condition including problem geometry.

The numerical solving of fluid flow in OpenFOAM was based on the simultaneous solution of the Navier-Stokes equations and the volume of fluid (VOF) method [12]. The volume of fluid method is a surface tracking technique [13]. The assumptions of this technique in the dam-break problem are:

(1)    Laminar flow (shallow water).
(2)    Over a flatbed.
(3)    No frictions.

The VOF method is based on scalar function, which is called friction function (C). The density term in the momentum equation is solved by following equation:

$$\frac{\partial C_i}{\partial t} + v \cdot \nabla C_i = 0 \tag{1}$$

In addition, we know the sum of the friction factor should be equal to one:

$$\sum_{i=1}^{n} C_i = 1 \tag{2}$$

At the end, the density ($\rho$) of fluid is calculated based on volume fraction:

$$\rho = \sum_{i=1}^{n} \rho_i C_i \tag{3}$$

For momentum conservation, the Navier-Stokes equation is presented as (note that the fluid is considered as Newtonian incompressible fluid):

$$\frac{\partial \rho U}{\partial t} + \nabla \cdot (\rho U U) - \nabla \cdot \mu \nabla U - \rho g = -\nabla p \tag{4}$$

where:

- $\frac{\partial \rho U}{\partial t}$ is the change of velocity with time;
- $\nabla \cdot (\rho U U)$ is the convective term;
- $\nabla \cdot \mu \nabla U$ is the velocity diffusion term;
- $\rho g$ is the body force term as external forces that act on the fluid (gravity);
- $\nabla p$ is the pressure term, fluid flows in the direction of largest change in pressure.

In addition, for mass conservation, the continuity equation is presented as:

$$\frac{\partial \rho}{\partial t} + \nabla \cdot (\rho U) = 0 \tag{5}$$

where $U$ is velocity, $\rho$ is density, $p$ is pressure, $\mu$ is dynamic viscosity, and $g$ is gravitational force. By involving the VOF technique and defining a new term called $\gamma$, which is phase fraction in one cell, the density would be ($\rho_m = \rho_1 \gamma_1 + \rho_2 \gamma_2$):

$$\rho_m = \gamma \rho_1 + (1 - \gamma) \rho_2 \tag{6}$$

For viscosity ($\mu_m = \mu_1 \gamma_1 + \mu_2 \gamma_2$):

$$\mu_m = \gamma \mu_1 + (1 - \gamma) \rho \mu_2 \tag{7}$$

where $\rho_m$ is mixture density, $\rho_1$ is water density, $\rho_2$ is air density, $\mu_m$ is mixture viscosity, $\mu_1$ is water viscosity, and $\mu_2$ is air viscosity.

$\gamma$ could be any value between one and zero, and this symbol represents the phase fraction in each computational cell; $\gamma = 1$ means that the cell is completely occupied by water, and if it is filled with air, $\gamma$ is zero. Generally, the phase fraction equation is:

$$\gamma = \frac{\text{fluid volume}}{\text{cell volume}} \tag{8}$$

Another equation is the transport equation (for property $\varphi$) following the general form:

$$\frac{\partial(\rho\varphi)}{\partial t} + \nabla\cdot(\rho\varphi U) - \nabla\cdot(\tau\,\nabla\varphi) = S_\varphi \tag{9}$$

where:

- $\frac{\partial(\rho\varphi)}{\partial t}$ represents the rate of change of property $\varphi$ with time;
- $\nabla\cdot(\rho\varphi U)$ is the advection of property $\varphi$ by the fluid flow (net rate of flow-convection);
- $\nabla\cdot(\tau\,\nabla\varphi)$ represents the rate of change due to diffusion of property $\varphi$ ($\tau$ is diffusion coefficient divided by the fluid density);
- $(S_\varphi)$ is the rate of change due to other sources ($\tau$ is diffusion coefficient divided by the fluid density).

The transport equation with the following form is used to calculate the relative volume fraction in each cell:

$$\frac{\partial\gamma}{\partial t} + \nabla\cdot(\gamma U) = 0 \tag{10}$$

By adding the artificial compression term into this equation, necessary compression of the surface is calculated:

$$\frac{\partial\gamma}{\partial t} + \nabla\cdot(\gamma U) + \nabla\cdot(\gamma\,(1-\gamma)\,U_r) = 0 \tag{11}$$

where $U_r$ is proper velocity field to compress the interface.

As mentioned, the velocity and pressure should be solved simultaneously. OpenFOAM has a specific algorithm called PIMPLE (PISO–SIMPLE) that was developed by merging pressure implicit split operator (PISO) and Semi-Implicit Method for Pressure-Linked Equations (SIMPLE) algorithms [14,15].

In Table 1, the values of fluid properties for each phase are shown.

**Table 1.** Fluid properties.

| Fluid Properties | Water (Phase 1) | Air (Phase 2) | Symbol | Unit |
|---|---|---|---|---|
| Kinematic viscosity | $1.0 \times 10^{-6}$ | $1.48 \times 10^{-5}$ | $\nu$ | $\text{m}^2{\cdot}\text{s}^{-1}$ |
| Density | $1.0 \times 10^{3}$ | $1.0$ | $\varrho$ | $\text{kg}{\cdot}\text{m}^{-3}$ |
| Surface tension | $0.07$ | $0.07$ | $\delta$ | $\text{N}{\cdot}\text{m}^{-1}$ |
| Velocity | - | - | U | $\text{m}{\cdot}\text{s}^{-1}$ |
| Pressure | - | - | p | $\text{N}{\cdot}\text{m}^{-2}$ |

## 4. Methodology

In this section, problem solving and specification of CFD datasets are discussed. Physical properties, tiering systems, grid classification, and boundary conditions are the topics covered in more detail. The method of the neural network, the number of inputs, and how the CFD model is defined in artificial intelligence are other sections of this topic. The workflow adopted was divided into two main phases—building the CFD model and NN model preparation (Figures 2 and 3).

Preparing a comprehensive dataset is one of the most significant parts of each data-mining (DM) project, and visualizing the data set is the first step of DM. The large dataset, which is generated by OpenFOAM, could be visualized and analyzed with the ParaView package (version 5.4, Kitware, Inc., Clifton Park, NY, USA). ParaView is an open-source software with the ability to visualize large datasets as well as convert the format of data. The OpenFOAM dataset included the computed variables for each grid at each time-step.

In this study, four variables were in need of prediction (phase fraction, pressure, x-direction velocity, and y-direction velocity). The number of variables of interest were based on the type of simulation and problem definition. The network needed to be trained for each variable separately

(four models). For training the model of each variable, 24 inputs were defined, including four variables for the cell of interest, four variables for the distance of the cell of interest from the boundaries, and four variables for each cell that was considered a tier (Table 2) [16].

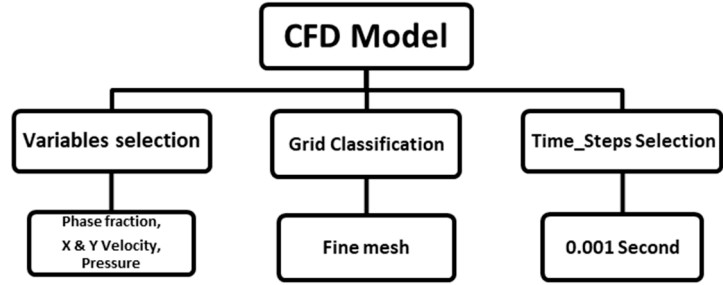

**Figure 2.** Computational fluid dynamic (CFD) model preparation workflow.

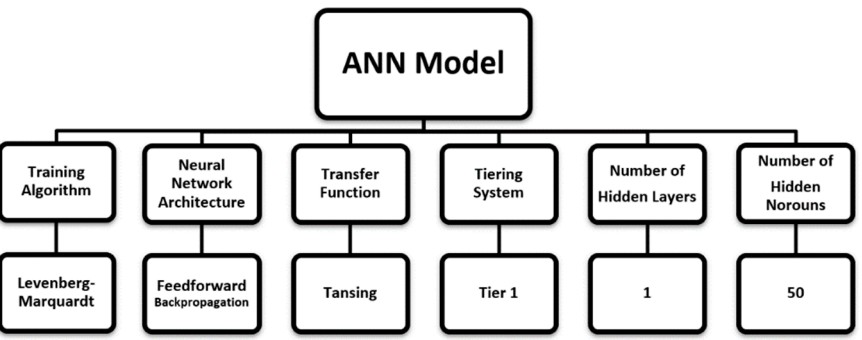

**Figure 3.** Artificial neural network (ANN) model preparation workflow.

**Table 2.** Input Details.

| | | |
|---|---|---|
| | 4 parameters from Main cell | Phase Fraction<br>x-direction velocity<br>y-direction velocity<br>pressure |
| Number of input: 24 | 16 parameters from tier cells | Phase Fraction<br>x-direction velocity<br>y-direction velocity<br>pressure |
| | 4 parameters for Location of main cell | distance 1<br>distance 2<br>distance 3<br>distance 4 |

The tiering system is a method of accounting for the neighboring cells. The information of the neighboring cells is considered as a feature in the NN input and includes pressure, velocities in x and y directions, and phase fraction.

This was a 2D problem, and as such, the four cells surrounding the cell of interest were accounted for as tier 1—one cell at the top, one at the bottom, one to the left, and one to the right side of the cell of interest. Tier 2 included the cells that shared a corner with the cell of interest, and tier 3 contained the cells in the second layer. In this research, only tier 1 data were counted (Figure 4) [17].

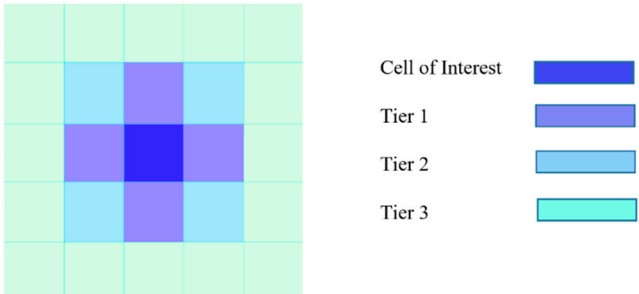

**Figure 4.** Schematics figure of tiering system.

Grid size selection depended on the numerical accuracy, resolution of the computed variables, and number of records generated. With the intention of achieving the best gridding mesh, three types of grid sizes were tested. First was the medium grid size of $50 \times 50$ (50 grids in x-axis and 50 grids in y-axis), second was a fine $200 \times 200$ grid, and the third was a very fine $400 \times 400$ grid. After running the simulation for all grid sizes, it was decided that fine classification ($200 \times 200$) was the appropriate size for this project (Figure 5). The medium grid size was rejected due to low regulation, and the very fine grid size was denied due to the increasing number of records. In Table 3, grid classification and related information are listed.

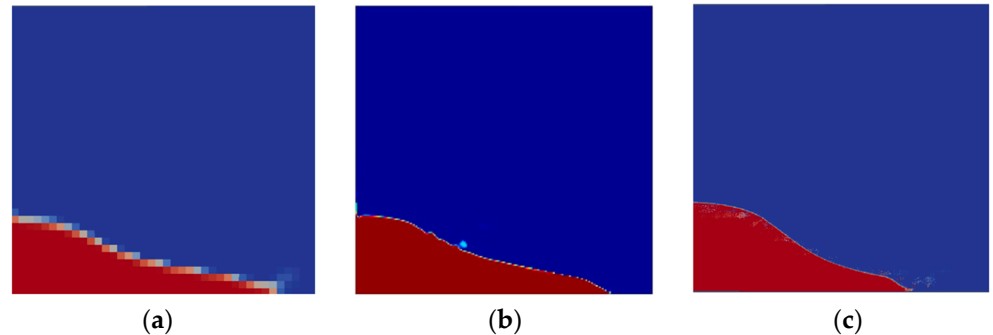

|     (a)     |     (b)     |     (c)     |

**Figure 5.** (**a**) Medium mesh with $50 \times 50$ grids, (**b**) fine mesh with $200 \times 200$ grids, (**c**) very fine mesh with $400 \times 400$ grids.

**Table 3.** Grid classification.

| Grid Classification | Dimension (x × y) | No. of Cells | Cell Size (cm) |
| --- | --- | --- | --- |
| Medium | $50 \times 50$ | 2500 | $1.168 \times 1.168$ |
| Fine | $200 \times 200$ | 40,000 | $0.292 \times 0.292$ |
| Very Fine | $400 \times 400$ | 160,000 | $0.146 \times 0.146$ |

Boundary conditions included four patches; three patches were walls located on the bottom, left, and right-hand side of the tank, and one patch was for the top, which was exposed to atmospheric conditions. Thus, both the outflow and the inflow were free to move up and down.

## 5. Artificial Neural Network (ANN)

The purpose of the neural network is to solve problems like a human brain does—with large interconnected neural units [18]. Essentially, training a NN model is choosing one model with minimum cost criterion through the set of generated models. Neural networks consist of multiple nodes. The input data are generated with nodes, and after doing simple mathematical operations, they transfer to other neurons. The output of each node is called "activation" or "node value". Each link between nodes is associated with a weight, and the neural network model could be learned by modifying weight values (Figure 6) [6].

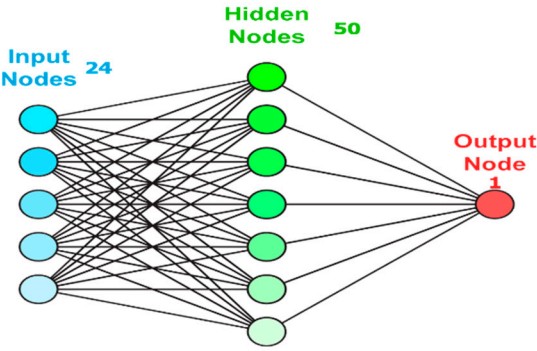

**Figure 6.** Schematic view of input, hidden layer, and output.

The NN model was built based on three sets of frameworks; first is the input data (predictor), second is the output data (target), and third is the intermediate layer called the hidden layer (Figure 7). When hidden layers are not applied while building a model, it is similar to linear regression, and when the hidden layers are included, it is non-linear regression [19].

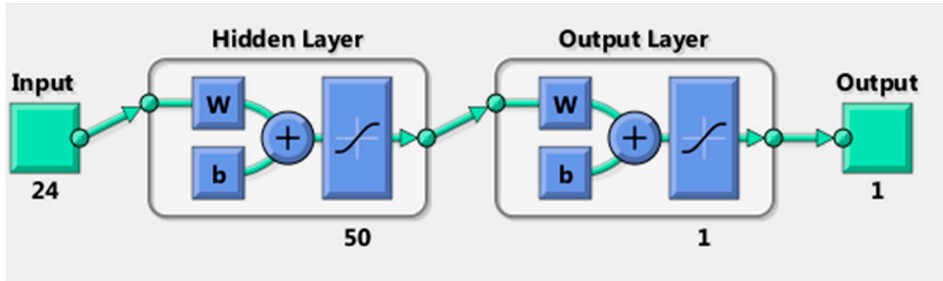

**Figure 7.** Input data, hidden layer, and output.

The number of inputs is defined based on the number of properties that are effective on target parameters. One hidden layer and 50 hidden neurons were selected for this problem. Many parameters needed to be considered when selecting the number of hidden neurons (number of input, number of output, number of samples in the training data set, physics of the problem, etc.). For this study, the number of hidden neurons was selected based on trial and error, and the best result was achieved using 50 hidden neurons. The last part of the model design is target definition (output) or variables that need to be predicted.

The Levenberg Marquardt algorithm (LMA) utilized in this study is based on the numerical solution with the purpose of minimizing non-linear functions. Interpolation between the GNA (Gauss–Newton algorithm) and the gradient descent method is the main strategy of LMA. This algorithm is fast and suitable for problems that include relatively small sizes of data [20].

The method of training used was feed-forward back propagation, which proceeds forward and computes the respective values. The computed values are then compared with the actual values, and based on the error value, it back propagates in order to reduce the error by changing and adjusting the weights [18].

Another important parameter in ANN is the transfer function that calculates output layers from the input. The transfer function takes the input value and returns it after converting between −1 and 1. For this model, tangent-sigmoid (Tansig) transfer function was selected [21,22]. "Tansig" is a non-linear transfer function that is typically used for feed-forward back propagation models in addition to imprinting complex features and learning the deep structure of the network. The use of the nonlinear transfer function is preferable to the linear transfer function (Figure 8).

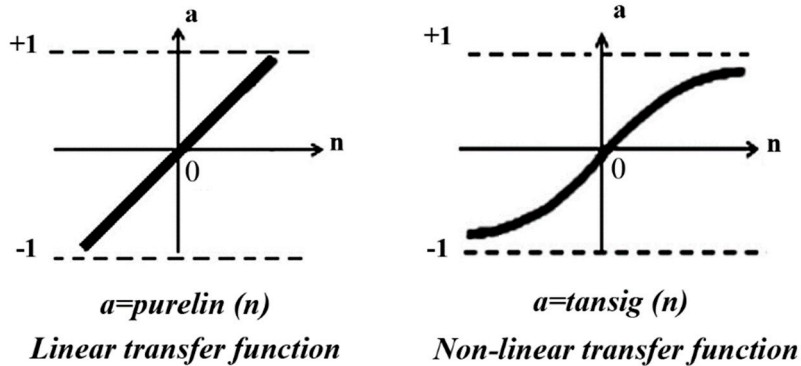

**Figure 8.** Transfer function.

## 6. Model Building and Deployment

Each network consists of two main sections, a building model and a deployment. Generally, the goal of each network is building a perfect model to predict properties. Determining the most effective variables is the first step of building the NN model.

The dependence of parameters on each other and the amount of dependency they display play a significant role in the input parameters determination. For building a comprehensive model, the user should consider all features that might be effective in problem approach.

Another important feature is selecting a reasonable temporal time-step in terms of dynamic changes. The temporal spacing between time-steps should be efficient—in other words, it should not be too long or too short because the system will not learn perfectly due to large or small dynamic changes.

Three samples of training performances for phase fraction, including temporal time-steps of 0.004, 0.002, and 0.001 s, are presented in Figures 9–11, and the corresponding models stopped after 3, 22, and 211, respectively, due to validation errors. Variables of time-step 575 were used as the input, and 576 was used as the output. The regression of the model with temporal time-step 0.001 was clearly improved compared to the other two models. Because the whole process was 3 s with 0.001 s as the temporal time-steps, the final model included 3000 time-steps.

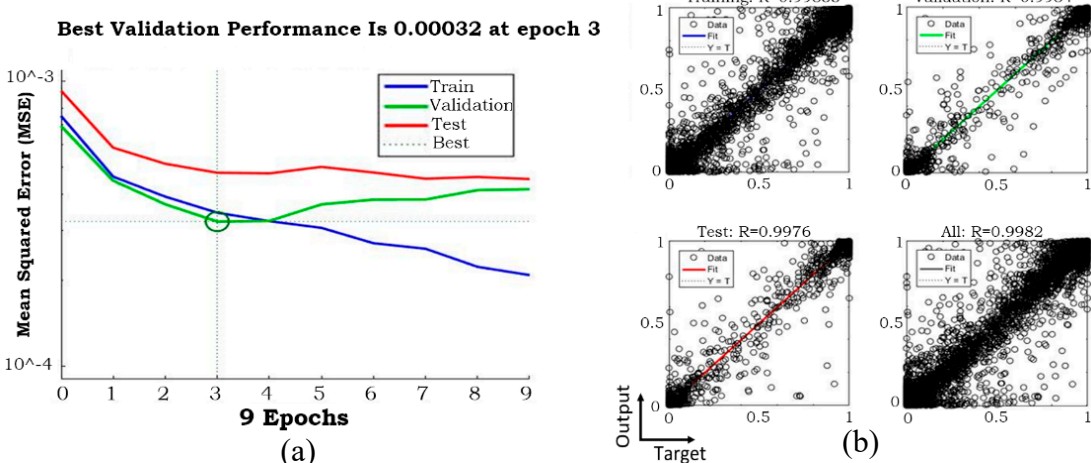

**Figure 9.** Neural network (NN) (**a**) validation performance and (**b**) regression when temporal time-step was 0.004 s.

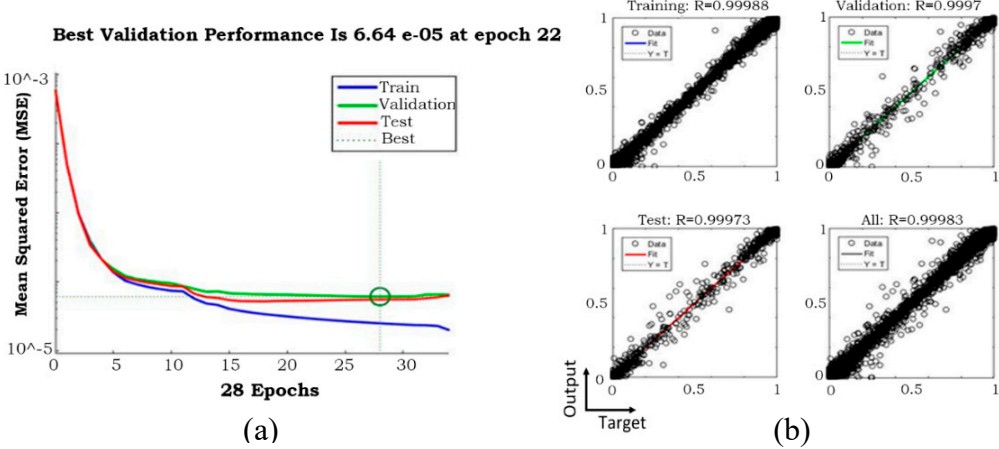

**Figure 10.** NN (**a**) validation performance and (**b**) regression when temporal time-step was 0.002 s.

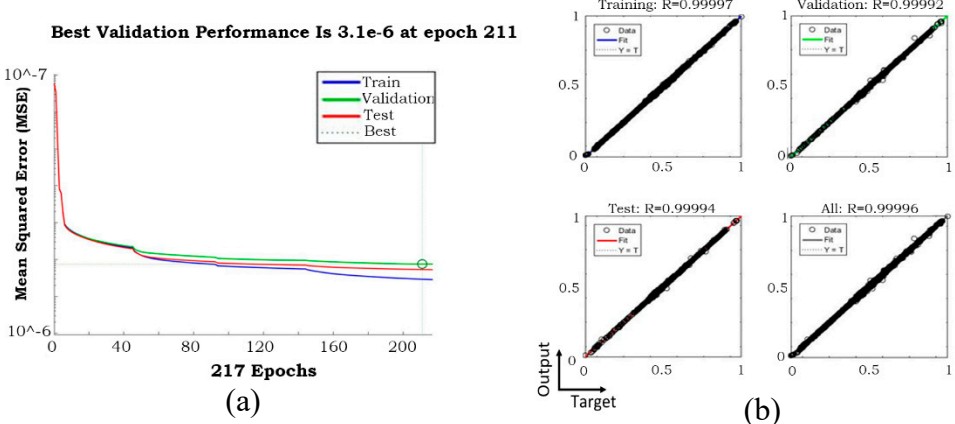

**Figure 11.** NN (**a**) validation performance and (**b**) regression when temporal time-step was 0.001 s.

By way of illustration, the results of the model with a 0.004 s temporal time-step indicate that this temporal time-step was too long for appropriate NN model training (Figures 12 and 13). The red points in the error distributions indicate at least a 10% error. Deployment started from time-step 1150, and the error percentage was not acceptable after two consecutive time-steps. Thus, 0.004 s as the temporal time-step was long and needed to be shorter.

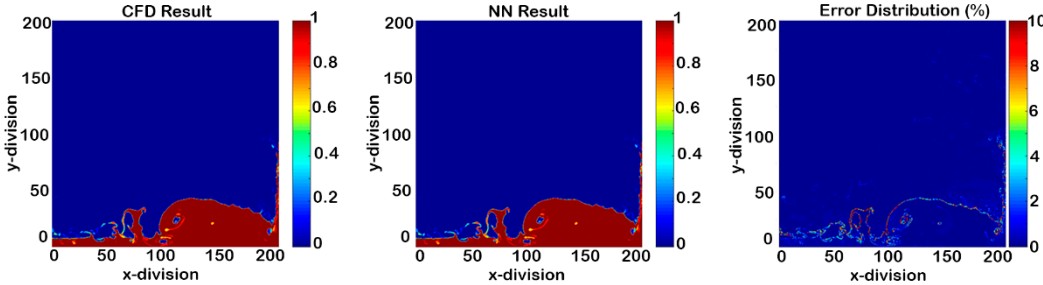

**Figure 12.** Phase fraction results in time-step 1150 (time-step size of 0.004 s).

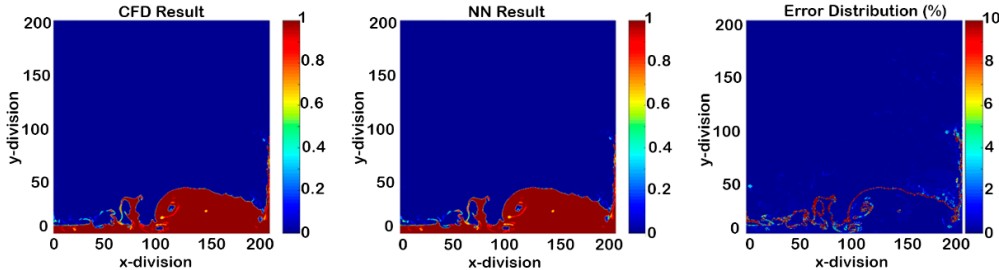

**Figure 13.** Phase fraction results in time-step 1151 (time-step size of 0.004 s).

The input consisted of all variables that were likely to affect the output. Twenty-four variables in time-step "n" were used as the input, and one variable from time-step "n + 1" was used as the output of the NN model. Because of the four interest variables, four models were needed (Figure 14).

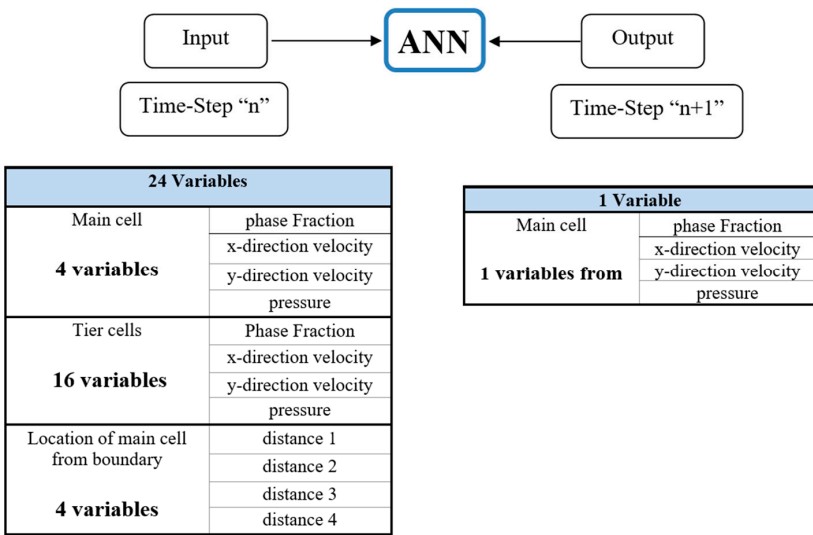

**Figure 14.** Number of output and input in ANN.

The network worked as a machine to find the trend between inputs and outputs. The machine received data and gave out a number based on its learning quality. Two types of data feed for the deployment of the model were provided—original data and output of NN, respectively called "non-cascading" and "cascading" conditions.

After the learning process and building the model, the deployment process began. The machine provided the output for each input dataset, and if the output of the machine was used as the input for the next time-step, it was called cascading (Figure 15). In the other words, by using one set of CFD data from one time-step, the rest of the time-step values could be predicted (the machine fed itself). Non-cascading referred to the conditions when the NN machine did not feed itself and used CFD dataset for each time-step (Figure 16). Building a comprehensive and perfect model of non-cascading was the first step in cascading deployment.

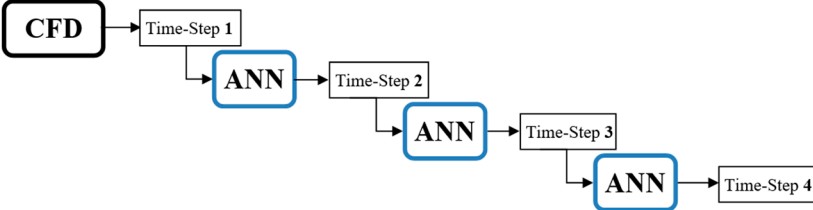

**Figure 15.** Cascading deployment process.

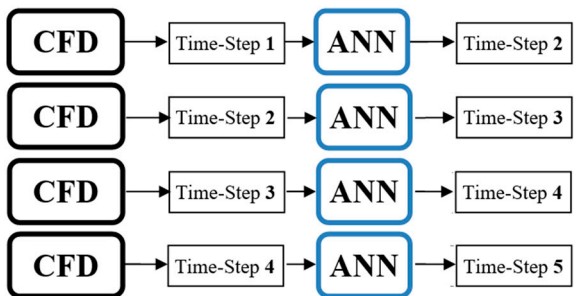

**Figure 16.** Non-cascading deployment process.

To understand and compare the error amounts of the NN and CFD models, appropriate errors needed to be defined for each parameter. For phase fraction, the error was defined as the absolute value of subtraction of CFD and NN.

$$Error = abs \left( \mathrm{CFD}_{\mathrm{output}} - \mathrm{NN}_{\mathrm{output}} \right) \tag{12}$$

For other parameters such as pressure and velocity, the values of CFD and NN needed to be normalized, and the error was defined as:

$$\mathrm{CFD}_{\mathrm{Norrmal}} = \frac{\left( \mathrm{CFD}_{\mathrm{output}} - \mathrm{Mean} \right)}{\left( \mathrm{MAX}_{\mathrm{CFD}_{\mathrm{output}}} - \mathrm{MIN}_{\mathrm{CFD}_{\mathrm{output}}} \right)} \tag{13}$$

$$\mathrm{NN}_{\mathrm{Normal}} = \frac{\left( \mathrm{NN}_{\mathrm{output}} - \mathrm{Mean} \right)}{\left( \mathrm{MAX}_{\mathrm{CFD}_{\mathrm{output}}} - \mathrm{MIN}_{\mathrm{CFD}_{\mathrm{output}}} \right)} \tag{14}$$

$$Error = abs \left( \mathrm{CFD}_{\mathrm{Norrmal}} - \mathrm{NN}_{\mathrm{Normal}} \right) \times 100 \tag{15}$$

There are several definitions for error calculation, but two of them are more prevalent for the assessment of NN performance. RMSE (root of mean square error) and MSE (mean square error) are commonly used to evaluate the performance of NN models. In this study, RMSE was utilized for the demonstration of error, as expressed below [23].

$$\mathrm{RMSE} = \sqrt{\frac{1}{n} \sum_{j=1}^{n} \left( y_{\mathrm{CFD}} - y_{\mathrm{NN}} \right)^2} \tag{16}$$

## 7. Results

In order to compare the results of the NN model prediction to the CFD data and subsequently visualize them, a 2D figure was generated for each time-step. Based on the grid values, each figure contained three subplots—the left side subplot showed OpenFOAM results, the middle subplot represented the prediction data made by the NN model, and the right side figure demonstrated the error percentage.

*Cascading and Non-Cascading Result*

Non-cascading results, which were presented previously, proved that the NN model had the capability to learn the entire process (3000 time-steps) using 30 time-steps. Thus, by selecting any time-step within the whole process and considering the CFD value as an input, the model was able to predict the next target with high accuracy. In other words, the input was coming from the CFD and the output was the next time-step predicted by ANN. However, in cascading deployment, just one time-step data from CFD was necessary and sufficient. Next, we examined what happened if the output of the NN model was used as an input of the next time-step. In this type of deployment,

there was no dependency on the CFD data for each time-step. Because the purpose of the cascading condition was the removal of any dependency on the CFD data, four general models were needed (phase fraction, pressure, x-direction velocity, and y-direction velocity) for deploying each parameter.

In the remainder of this section, four ideas of cascading condition are compared with their results.

(1) Model I—using 16 time-steps data from significant dynamics movements

In order to have a comprehensive model for the entire process, all variables from the most significant dynamic movements were selected for the training procedure (Figure 17).

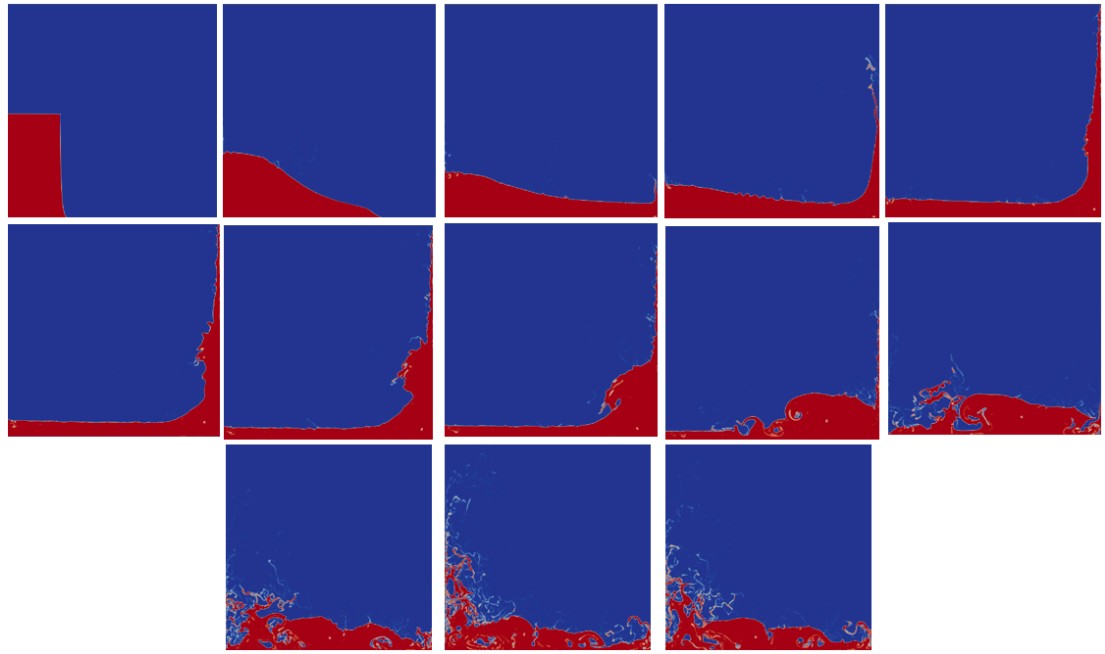

**Figure 17.** Significant dynamics selected for the training process.

For Model I, it was concluded that using significant dynamic movements was a great enhancement in order to have a perfect model for the entire process. Nevertheless, in some time-steps, especially in pressure and velocity, noticeable errors were observed and are presented below in Figures 18–20. The RMSE graph of the cascading condition is shown in Figure 21 and indicates the ability of the model to predict the value of each parameter up to six time-steps after deployment. As a means of improving accuracy, more time-steps from the parts where the most errors occurred were included (Model II).

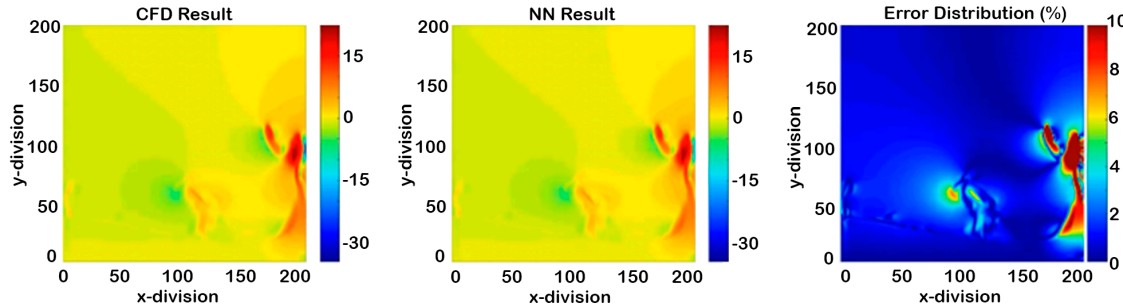

**Figure 18.** Non-cascading result of *Y*-direction velocity at the maximum error in the whole process (2.604 s).

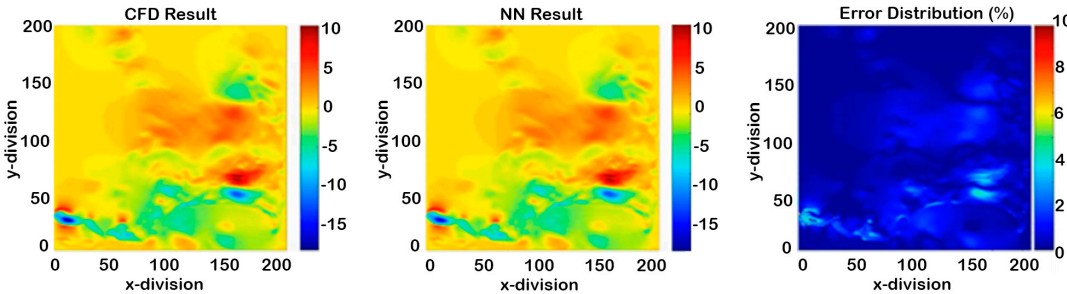

**Figure 19.** Non-cascading result of X-direction velocity at the maximum error in the whole process (2.604 s).

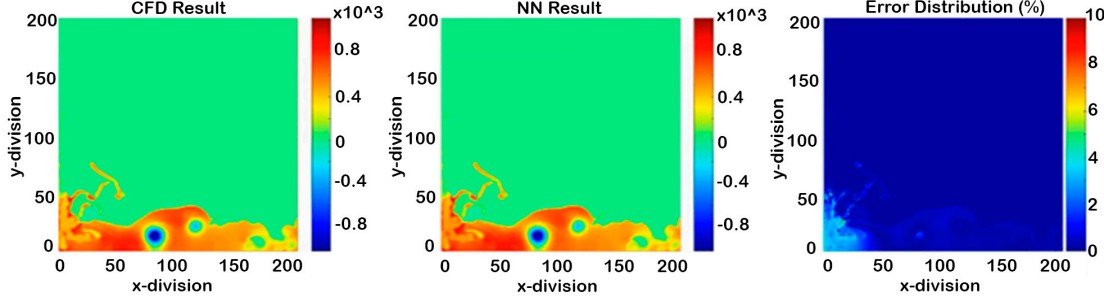

**Figure 20.** Non-cascading result of pressure at the maximum error in the whole process (2.604 s).

For starting the deployment of the cascading condition, all of the variables from CFD could be used. To have the same condition for all models, we started the deployment from time-step 1000 to see how many time-steps would be predicted.

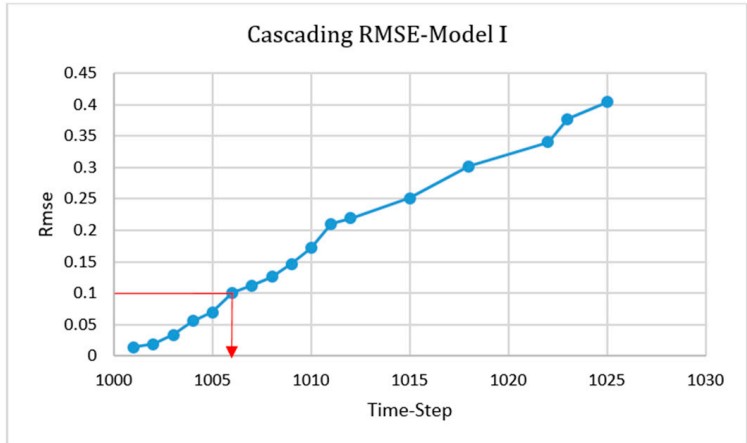

**Figure 21.** Root mean square equation (RMSE) of the cascading condition for Model I.

(2) Model II—involving time-steps with highest error in training system

A new model for each parameter was built by involving high error time-steps. This consideration improved the non-cascading model. In a previous section, 16 time-steps were used for the NN training process, but this number was increased to 30 for each model. The cascading condition results of this model after the first errors appeared are shown in Figures 22–25, and the related RMSE graph is shown in Figure 26. It indicates that the model was able to predict the value of each parameter up to nine time-steps after deployment.

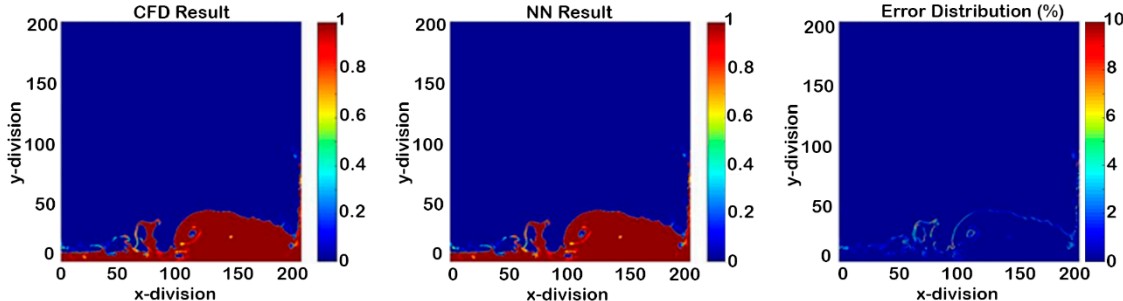

**Figure 22.** Cascading results of Model II for phase fraction (seven time-steps after deployment).

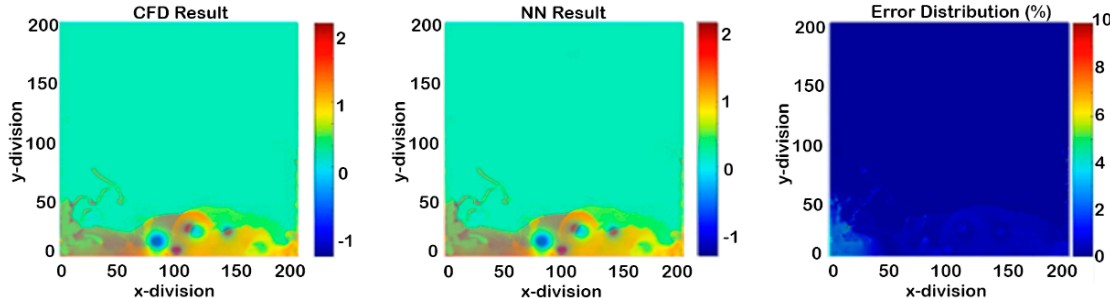

**Figure 23.** Cascading results of Model II for pressure (seven time-steps after deployment).

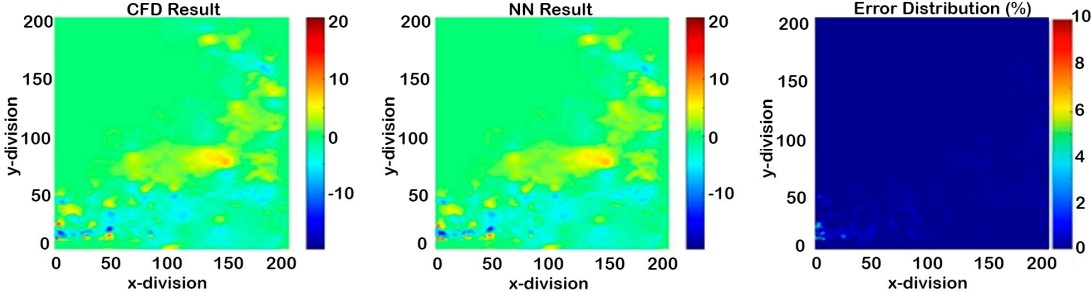

**Figure 24.** Cascading results of Model II for *X*-direction velocity (seven time-steps after deployment).

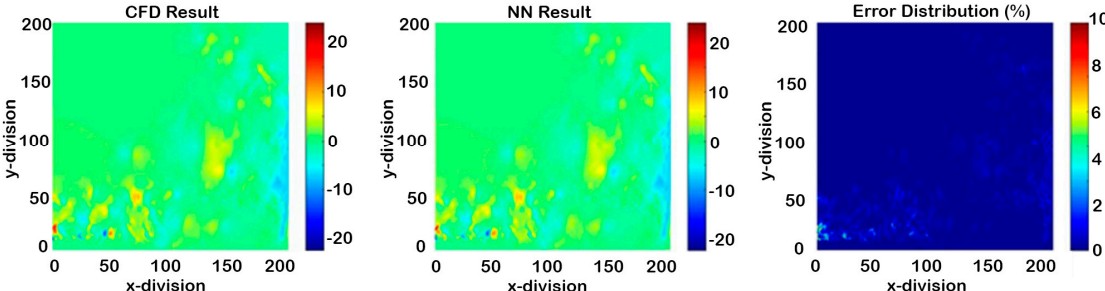

**Figure 25.** Cascading results of Model II for *Y*-direction velocity (seven time-steps after deployment).

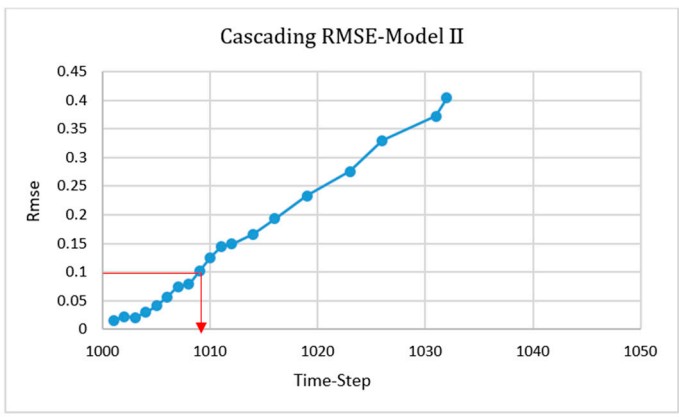

**Figure 26.** Root mean square equation (RMSE) of the cascading condition for Model II.

(3) Model III—Reduction of data

By looking at the CFD results, it was found that values in many grids did not change significantly over the large number of time-steps. One of the ideas that was performed in order to increase the accuracy of model was changing the distribution of data. For instance, the majority of phase fraction values remained constant from the beginning to the end of the simulation, and they were either zero or one. By eliminating these data, a new model was trained with smaller amounts of data. The percentage of eliminated data for each parameter is expressed in Table 4.

The black shadows below indicate eliminated grids, which were approximately constant during entire process (Figure 27).

**Table 4.** Percentage of eliminated data for each parameter.

| Parameter | Number of Total Grids | Number of Constant (Eliminated) Grids | Number of Non-Stationary Grids | % of Elimination |
|---|---|---|---|---|
| Phase fraction | 40,000 | 13,327 | 26,673 | 33% |
| Pressure | 40,000 | 17,235 | 22,765 | 43% |
| X-direction-velocity | 40,000 | 10,336 | 29,664 | 26% |
| Y-direction-velocity | 40,000 | 10,214 | 29,786 | 25% |

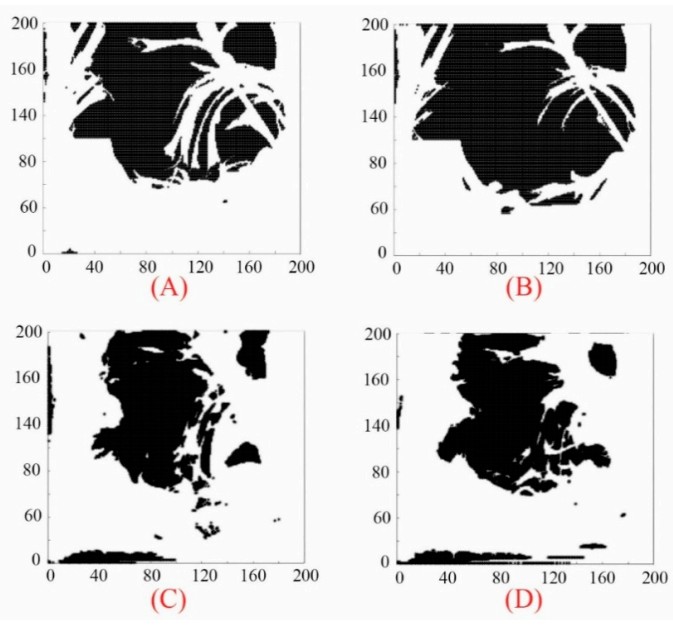

**Figure 27.** Eliminated grids for (**A**) phase fraction, (**B**) pressure, (**C**) *x*-direction-velocity, (**D**) *y*-direction-velocity.

Data distribution was modified in order to reduce the negative effects of redundant data. The histogram of data for each parameter before and after elimination are shown in Figures 28–31. It is clear from the histogram of the phase fraction that the frequency of "zero" and "one" decreased significantly after elimination. For the other parameters, most of the grids value were zero, and after removing redundant data, the frequency decreased to a reasonable amount.

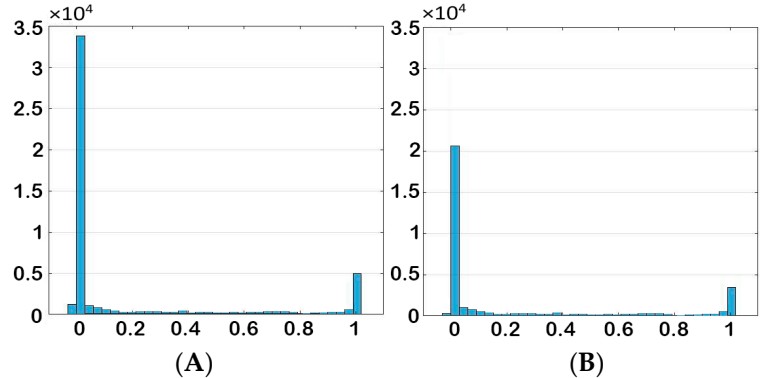

**Figure 28.** Histogram of phase fraction (**A**) before and (**B**) after elimination.

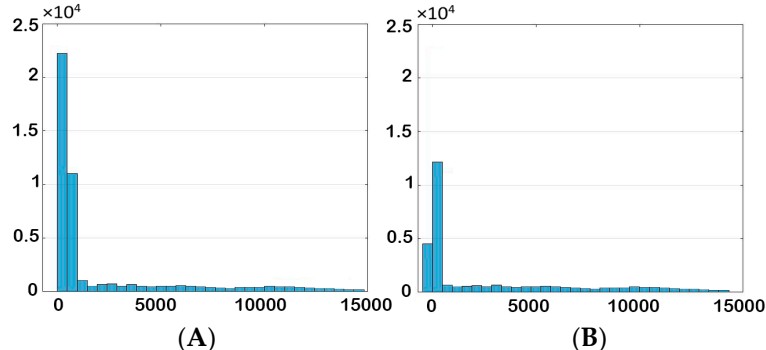

**Figure 29.** Histogram of pressure (**A**) before and (**B**) after elimination.

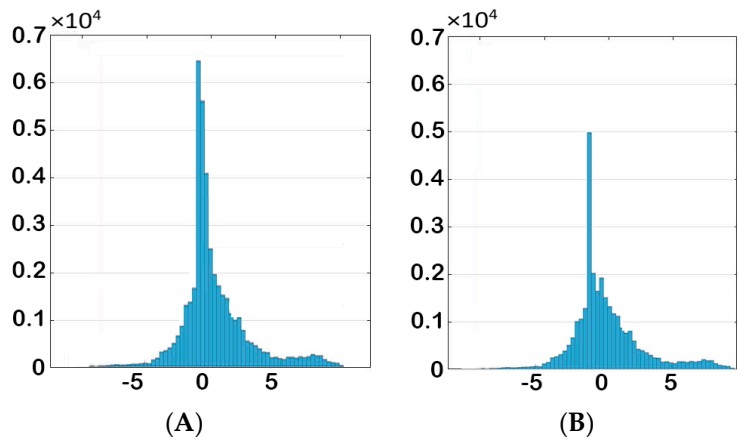

**Figure 30.** Histogram of *y*-direction-velocity (**A**) before and (**B**) after elimination.

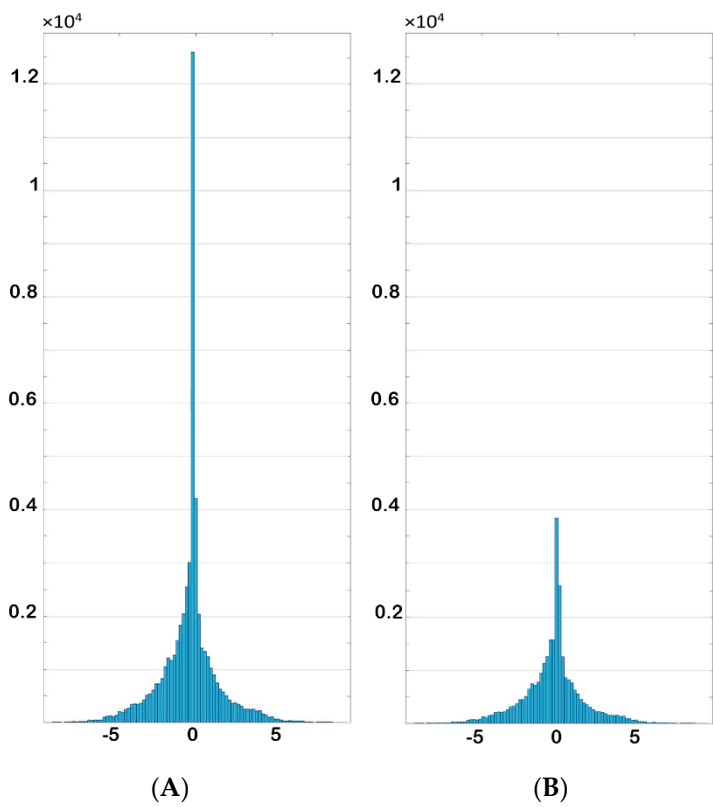

**Figure 31.** Histogram of *x*-direction-velocity (**A**) before and (**B**) after elimination.

The cascading condition results of Model III once the first errors appeared are shown in Figures 32–35, and the RMSE graph is shown in Figure 36 and indicates the ability of the model to predict each parameter's value up to fifteen time-steps after deployment.

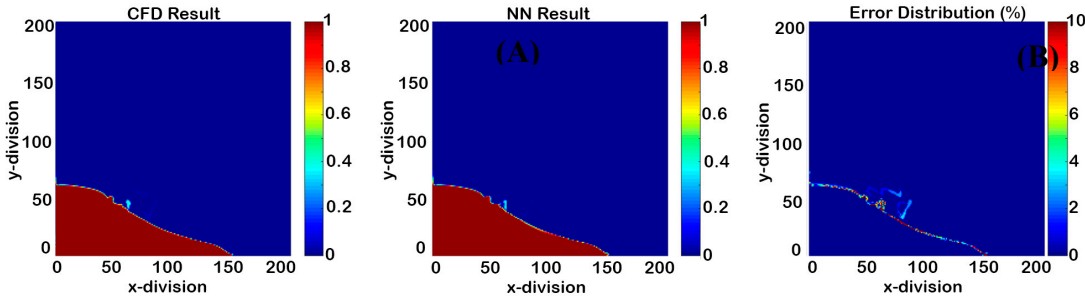

**Figure 32.** Cascading results of Model III for phase fraction (14 time-steps after deployment).

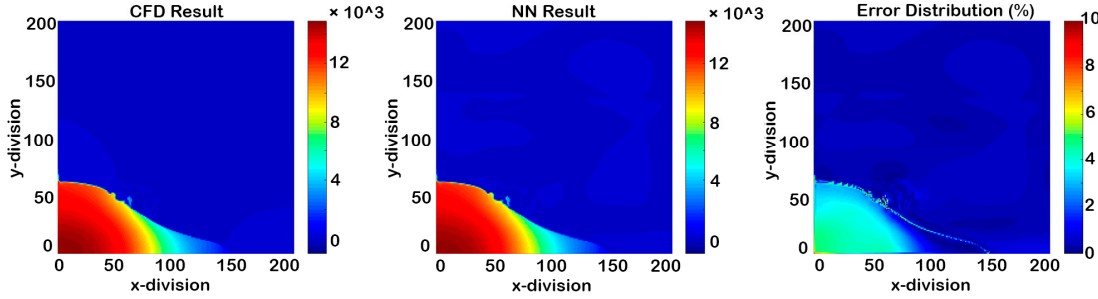

**Figure 33.** Cascading results of Model III for pressure (14 time-steps after deployment).

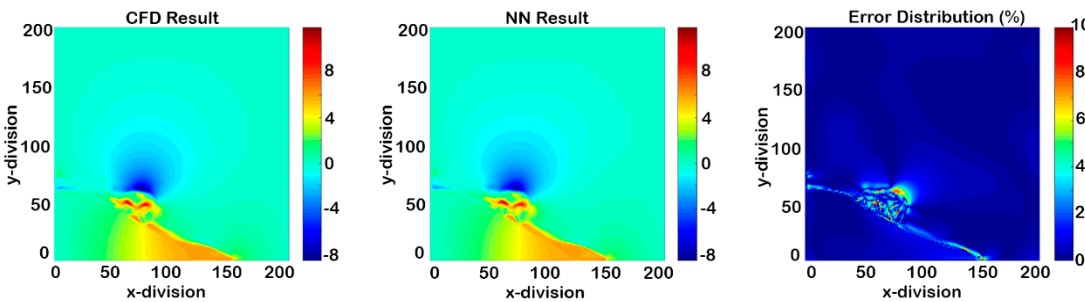

**Figure 34.** Cascading results of Model III for *X*-direction-velocity (14 time-steps after deployment).

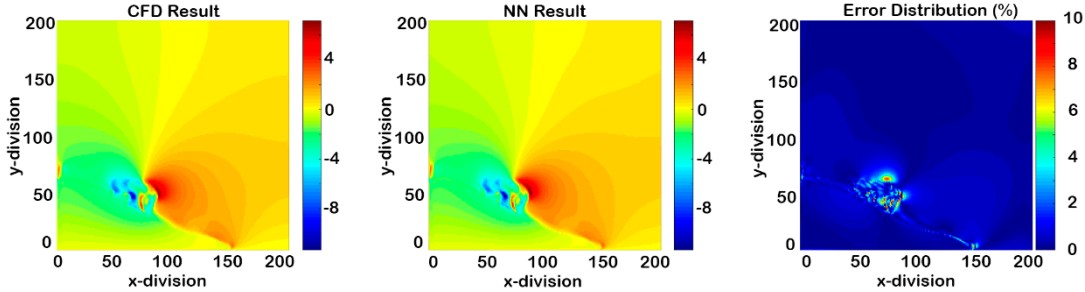

**Figure 35.** Cascading results of Model III for *Y*-direction-velocity (14 time-steps after deployment).

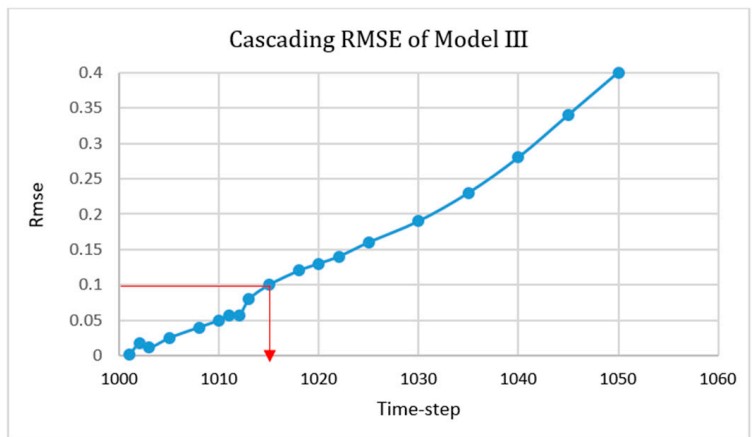

**Figure 36.** Root mean square equation (RMSE) of the cascading condition for Model III.

(4) Model IV—Building a model based on 200 time-steps

This model was a combination of Model II and Model III. It focused on 200 time-steps instead of entire process in order to achieve a better non-cascading model. Therefore, in every 20 time-steps from time-step 1000 to 1200, constant grids were eliminated. Figures 37–40 represent the results after 50 time-steps, and red dots indicate the 10% error. The accuracy of this SPM was more than 90% in order to predict the flow parameters of the next 45 time-steps. The RMSE graph of the cascading condition is shown in Figure 41.

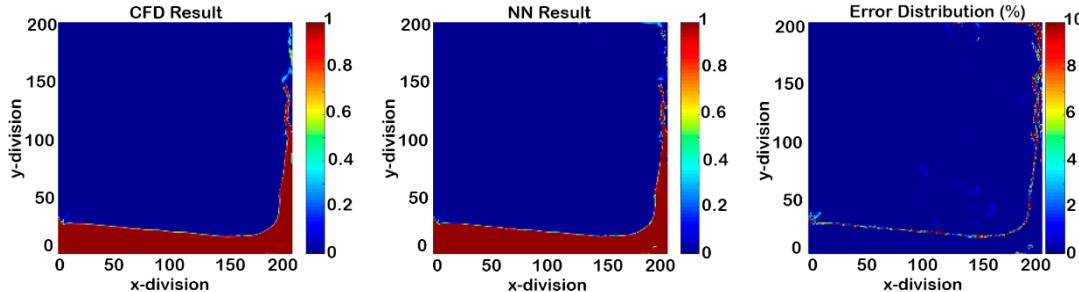

**Figure 37.** Cascading results of Model IV for phase fraction at time-step 1050 (50 time-steps after deployment).

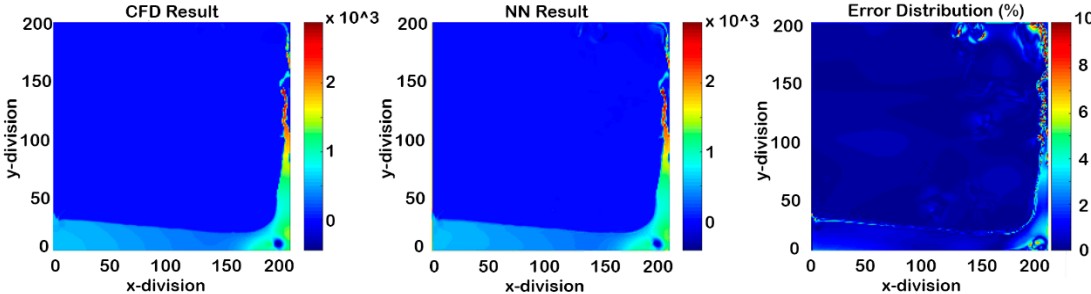

**Figure 38.** Cascading results of Model IV for pressure at time-step 1050 (50 time-steps after deployment).

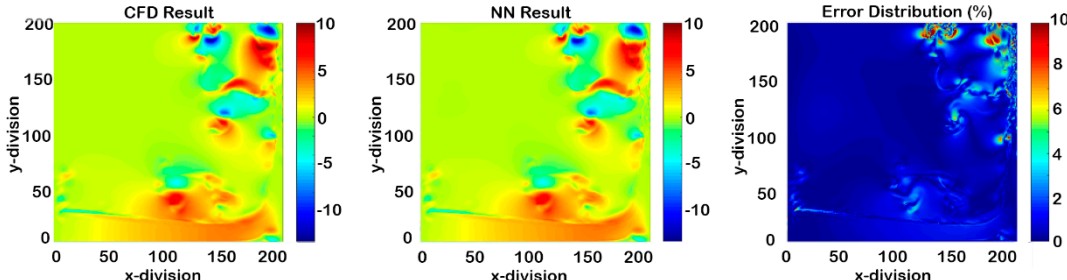

**Figure 39.** Cascading results of Model IV for X-direction-velocity at time-step 1050 (50 time-steps after deployment).

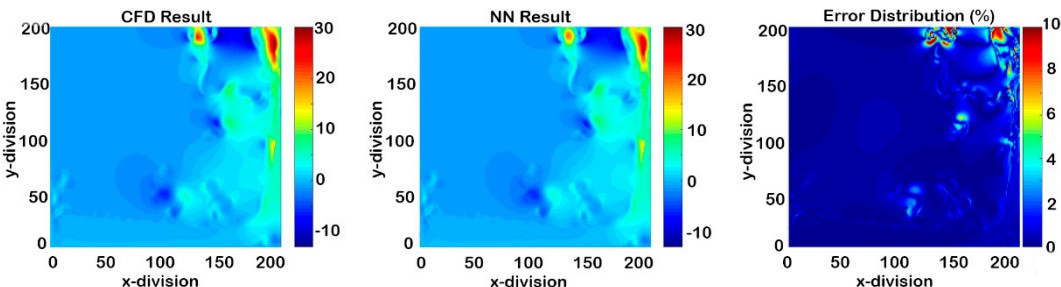

**Figure 40.** Cascading results of Model IV for Y-direction-velocity at time-step 1050 (50 time-steps after deployment).

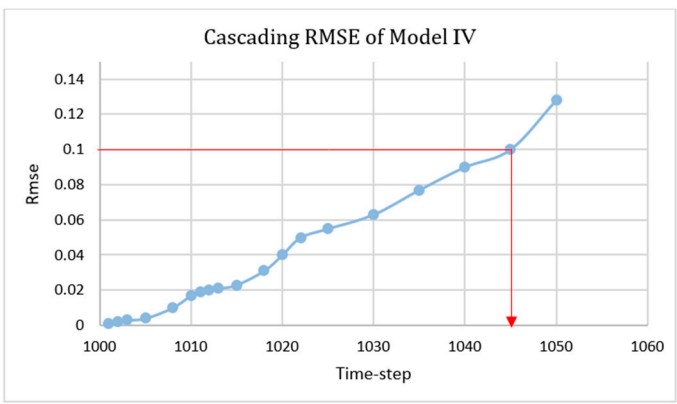

**Figure 41.** Root mean square equation (RMSE) of the cascading condition for Model IV.

## 8. Conclusions

The main objective for this work was to predict the physical parameters such as velocity, pressure, and phase fraction during the process, along with reducing the computational time. The dam-break problem, which is one of the most well-known CFD problems, was selected in order to examine whether the neural network was capable of predicting the properties. The OpenFOAM package was used for simulating the problem and generating the data for building the neural network model. After performing several ideas, the last model was designed based on 24 parameters and 40,000 grids.

Two scenarios were considered; the first was non-cascading, which meant building a model for each property that was able to predict the results at any time-step using the information from the prior time-step. The second scenario was cascading, which meant building a model that was able to use CFD results in a time-step and predict several following time-steps.

For the first scenario, the results indicated that the neural network model was able to predict the results with high accuracy and was perfectly matched with the CFD results. For the cascading, the model was able to predict up to six time-steps for Model I, nine time-steps for Model II, 15 time-steps for Model III, and 45 time-steps for Model IV with an acceptable error percentage (<%10) (Figure 42).

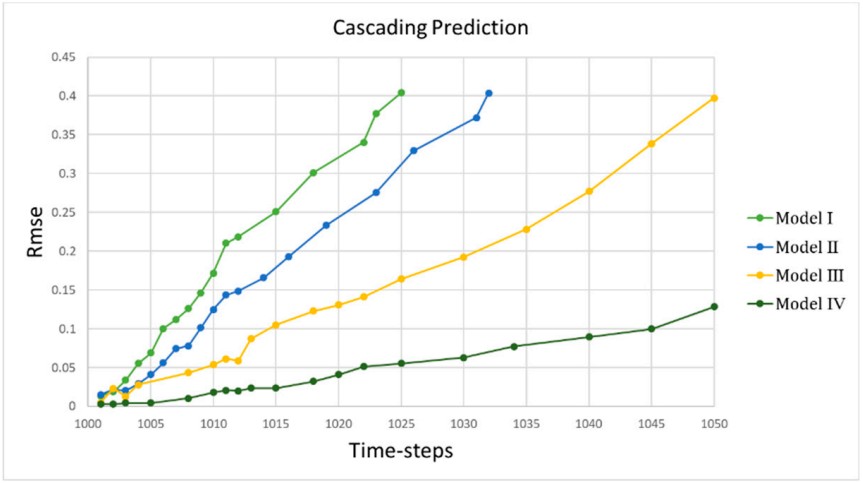

**Figure 42.** Comparison of the results.

Reducing the time process compared to CFD was the main objective of the research, thus this process took 8 h with four processors when it was done by CFD and took 2 min by smart proxy with one processor. A noticeable impact in the application of AI for the prediction of the dynamic parameters could be observed in the enhanced shale oil and gas recovery and $Co_2$ sequestration [24–26].

Following this study's context, focusing more on cascading conditions, applying the model with different fluid properties, and building 3-D models that include three phases, are future studies of this project.

**Funding:** This research received no external funding.

**Conflicts of Interest:** The author declare no conflict of interest.

**Nomenclature**

| | |
|---|---|
| $\rho$ | Density |
| $\mu$ | Dynamic viscosity |
| $\sigma$ | Surface tension coefficient |
| $g$ | Gravitational force |
| $k$ | Surface curvature |
| $\gamma$ | Second viscosity |
| $U$ | Velocity |
| $\eta$ | Dynamic |
| $P$ | Pressure |
| $\tau$ | Diffusion coefficient over density |
| $\alpha$ | Phase fraction |
| VOF | Volume of Fluid |
| RANS | Reynolds-Averaged Navier-Stokes |
| CFD | Computational fluid dynamics |
| AI | Artificial intelligence |
| NN | Neural network |
| ANN | Artificial neural network |
| SPM | Smart proxy model |
| RMSE | Root Mean Square Error |

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
