# Peer review of "Predicting the Dynamic Parameters of Multiphase Flow in CFD (Dam-Break Simulation) Using Artificial Intelligence-(Cascading Deployment)"

_fluids, doi:10.3390/fluids4010044_

Round 1
Reviewer 1 Report
The paper develops a model which help all engineering aspects of oil and gas industry from drilling, well design to future prediction of an efficient production. This is the cool achievement about this work.
Would like to suggest authors to cite the following four manuscripts which are also highlighting the great potential of ANN models in petroleum industry, especially as proxy models:
1) Amirian, E., Leung, J. Y., Zanon, S., & Dzurman, P. (2015). Integrated cluster analysis and artificial neural network modeling for steam-assisted gravity drainage performance prediction in heterogeneous reservoirs. Expert Systems with Applications, 42(2), 723-740.
2) Amirian, E., Fedutenko, E., Yang, C., Chen, Z., & Nghiem, L. (2018). Artificial Neural Network Modeling and Forecasting of Oil Reservoir Performance. In Applications of Data Management and Analysis (pp. 43-67). Springer, Cham.
3) Fedutenko,
E., Yang, C., Card, C., & Nghiem, L. X. (2014, June 10).
Time-Dependent Neural Network Based Proxy Modeling of SAGD Process.
Society of Petroleum Engineers. doi:10.2118/170085-MS
4) Mohaghegh, S. D. (2018). Smart Proxy Modeling. In Data-Driven Analytics for the Geological Storage of CO2 (pp. 33-44). CRC Press.
Author Response
I appreciate your kind words about this article and thank you for your comments.
The text has been updated as suggested. (Page 2, 21 and 22)

Reviewer 2 Report
This manuscript has introduced a fast-track data-driven method based on Artificial Intelligence (AI) for prediction of multiphase flow. The paper is well written. I recommend the paper to be published in a revised version.
Please clarify the questions below:
1. As seen in Fig. 14, the input-output pair is defined for one main cell and each variable has its own model. In the test cases, there are 40,000 cells. Do you repeat the training procedure (in Fig.14) for each cell?
2. The training data set was selected at 15 or 30 time steps within 3000 time steps. Will the model be able to predict the variables beyond 3000 time steps?
3. What is the CPU time for models I-IV?
Author Response
We sincerely thank the reviewer for positive feedback and valuable comments.
Our responses to the reviewer ’s comments are given below:
1. As seen in Fig. 14, the input-output pair is defined for one main cell and each variable has its own model. In the test cases, there are 40,000 cells. Do you repeat the training procedure (in Fig.14) for each cell?
Reply: As you mentioned, the number of cells are 40000 (200 cells in x and 200 cells in y directions). For building the model, these 40000 cells will be the number of the rows and 24 parameters will be number of the columns in input matrix. on the other hand the output matrix has just one column with 40000 rows. Since there are 4 variables, 4 for different models need to be built. The input matrix is same for all these models but the output is dependent to the variable which is needed to be predict.
2. The training data set was selected at 15 or 30 time steps within 3000 time steps. Will the model be able to predict the variables beyond 3000 time steps?
Reply: Thank you for your question. Actually this is one the objective of cascading deployment that the model be able to predict prior and further time-steps. For this problem, our model was capable to predict the behavior of the fluid up to around 40 time-steps after deployment with high accuracy. For example if we start the deploying process from time-step 3000, the ANN model can predict the parameters up to 30040.
3. What is the CPU time for models I-IV?
Running the CFD simulation using four CPUs took 8 hours. For building the ANN model, the time of training processes were different and dependent on number of the rows, for example the number of rows for input matrix in model Ⅰ and II were 16×40000=640,000 and 30×40000=1,200,000 respectively. But deploying process for all models were 100 to 120 seconds.

Reviewer 3 Report
Line 11, please change “is” to “are”;
Line 14, “Unconventional” to “unconventional”;
Line 56, “Authors” to “authors”;
There are too many errors in the entire manuscript;
I doubt the ANN method works for CFD simulations. If a different model is chosen, how can you predict the results based the model trained in this paper?
The author just combined concepts from different areas, such as oil, gas, and multiphase flow. The manuscript only studied water.
Author Response
Thank you for your time to read this article. Please see below, our detailed response to the comments.
The typos are corrected in text.
Comment: I doubt the ANN method works for CFD simulations. If a different model is chosen, how can you predict the results based the model trained in this paper?
Reply: Please refer to the latest articles with CFD-ANN combination, you will find out the progress made in this area.
Regarding your concern about the performance of the model presented in this paper in predicting the parameters of different model, it all depends on what you mean by different models. Do you mean different geometries? Or, different properties? Or different problems all together?
Anyways, is it possible to use one CFD model for simulating or predicting the parameters of all CFD problems in the world? Obviously, each single problem needs to be designed and built based on the problem characteristics.
So, the answer to your questions is: Yes. As long as by “different problem” you mean modification of different characteristics of the same CFD model, then yes, this AI-based technique can handle it.
Comment: The author just combined concepts from different areas, such as oil, gas, and multiphase flow. The manuscript only studied water.
Reply: The title of this article is “Predicting the Dynamic Parameters of Multiphase Flow in CFD (Dam-Break Simulation) Using Artificial Intelligence- (Cascading Deployment), and we mentioned the word “Dam-Break” clearly. We chose “Dam-Break” problem because of its prominence in the CFD field. In addition, if you pay attention to the “Table 1”, as is written “Phase 1” and “Phase 2”, the fluid properties can be changed to the properties of any fluid.
Another thing, in our problem, one of the main dynamic parameters is phase fraction, the percentage of liquid phase in each grid (For phase fraction definition please refer to page 3), hence the problem is multiphase flow including liquid and gas.
Round 2
Reviewer 2 Report
I am happy with the responses and the current version. Suggest it be accepted.